# Electroporation as the Immunotherapy Strategy for Cancer in Veterinary Medicine: State of the Art in Latin America

**DOI:** 10.3390/vaccines8030537

**Published:** 2020-09-17

**Authors:** Felipe Maglietti, Matías Tellado, Mariangela De Robertis, Sebastián Michinski, Juan Fernández, Emanuela Signori, Guillermo Marshall

**Affiliations:** 1Instituto Universitario del Hospital Italiano de Buenos Aires, CONICET, Buenos Aires 1199, Argentina; 2VetOncologia, Veterinary Oncology Clinic, Buenos Aires 1408, Argentina; vetoncologia@gmail.com (M.T.); juanmfer@yahoo.com (J.F.); 3CNR-Institute of Biomembrane, Bioenergetics, and Molecular Biotechnology, 70126 Bari, Italy; m.derobertis@ibiom.cnr.it; 4Department of Bioscience, Biotechnology, and Biopharmaceutics, University of Bari, 70126 Bari, Italy; 5Instituto de Física del Plasma, DF, FCEyN, UBA-CONICET, Buenos Aires 1428, Argentina; sebastianmichi@gmail.com (S.M.); marshall.guillermo@gmail.com (G.M.); 6Laboratory of Molecular Pathology and Experimental Oncology, Institute of Translational Pharmacology, CNR, 00133 Rome, Italy; emanuela.signori@ift.cnr.it

**Keywords:** gene electrotransfer, electrochemotherapy, cancer, tumor, immune response, gene therapy, companion animals

## Abstract

Electroporation is a technology that increases cell membrane permeability by the application of electric pulses. Electrochemotherapy (ECT), the best-known application of electroporation, is a very effective local treatment for tumors of any histology in human and veterinary medicine. It induces a local yet robust immune response that is responsible for its high effectiveness. Gene electrotransfer (GET), used in research to produce a systemic immune response against cancer, is another electroporation-based treatment that is very appealing for its effectiveness, low cost, and simplicity. In this review, we present the immune effect of electroporation-based treatments and analyze the results of the vast majority of the published papers related to immune response enhancement by gene electrotransfer in companion animals with spontaneous tumors. In addition, we present a brief history of the initial steps and the state of the art of the electroporation-based treatments in Latin America. They have the potential to become an essential form of immunotherapy in the region. This review gives insight into the subject and helps to choose promising research lines for future work; it also helps to select the adequate treatment parameters for performing a successful application of this technology.

## 1. Introduction

Veterinary medicine has been continually growing as owners seek for better care for their pets. Companion animals have become a vital component of many families, and the expenditure in their treatments has been increasing steadily in the last decades [1]. In the past, veterinarians used to treat every disease without distinction, but in recent times some of them have specialized in specific fields, giving rise to different medical specialties in veterinary medicine such as the ones which exist in human medicine. Veterinary oncology nowadays is a very developed field, which in many countries, is even more advanced than its human counterpart. For example, in Latin America, Electrochemotherapy (ECT) is available in many countries in veterinary medicine but not in human medicine.

ECT is a treatment for cancer which consists in the application of an electric field to increase cell membrane permeability to bleomycin or cisplatin. It was first described by Mir et al. in 1991 [2], and the first clinical trial was published later that same year [3]. In 2006 when the standard operating procedures for the technique were published, and the Cliniporator (IGEA, Carpi, Italy) was approved for human use, it became a valuable option for palliative treatment of cancer patients.

Before using electric fields to introduce cisplatin or bleomycin into cells, they were used to introduce genes and increase their expression using a technique called Gene Electrotransfer (GET). It was first introduced by Neumann et al. in 1982 [4], who described it as a simple, easily applicable, and efficient method for gene transfection. Despite its earlier discovery, GET is still an experimental therapy, while ECT is a standard of care for selected patients, both in human and veterinary medicine.

Recently, the development of calcium electroporation has allowed the replacement of cisplatin or bleomycin with calcium. Its main indication is when it is not possible to use the previous drugs, but although it preserves selectivity, it is less effective [5].

These techniques share a common characteristic, they provoke a transient formation of pores in the cell membrane, short enough to spare the cells. On the contrary, irreversible electroporation (IRE), another electroporation-based technique, induces long pore resealing times, leading to cell death by irrecoverable damage to its homeostasis [6]. Performing reversible or irreversible electroporation depends on the electrical pulse parameters applied. 

The high response rate of electroporation-based treatments for biomedical applications made them an increasingly-used therapeutic procedure in oncology both in human and veterinary medicine. Although the treatments are rather simple in all cases, their success depends on several factors: First, the type of electrodes used and its configuration with adequate electrical parameters depending on which electroporation-based treatment is performed, i.e., amplitude, duration, and the number of electric pulses. Second, the correct use of the electrodes to cover the tumor with an adequate electric field [6,7,8,9]. Third, a sufficient concentration of the drug in the site of treatment (except for IRE) [10]. Fourth, the immunological response of the host. In all electroporation-based therapies for cancer, the immune system plays a crucial role in the response. This technology may be the key to the development of highly effective immunotherapy in a more affordable way [11,12]. 

Here we will explore the reasons why each of the electroporation-based treatments mentioned could be a good candidate for developing an immunogenic strategy, the state of the art of electroporation-based treatments, and the availability of electroporation devices for their application in Latin America.

## 2. Irreversible Electroporation

IRE induces cell death by irrecoverable structural changes in the cell membrane, provoking the loss of cell homeostasis [13]. IRE-based therapeutic approaches consist of a local ablative treatment produced by the administration of at least 80–100 pulses of high amplitude (from 1000 to 3000 V/cm, with currents up to 40 A, depending on tissue electrical resistance). This method has found extensive application as a non-thermal form of tumor ablation in human and veterinary medicine [14,15,16,17,18], even for intra-abdominal malignancies [19,20], and endovascular cardiac ablation [21].

Although IRE is not selective for cancer cells, it induces immunogenic cell death, activates dendritic cells, and alleviates stroma-induced immunosuppression without affecting the tumor-restraining collagen [22]. IRE treatments performed in immunodeficient mice showed a significantly poorer response. The re-challenge by the injection of tumoral cells provoked new tumors in these mice, while not in the immunocompetent ones [11]. 

## 3. Reversible Electroporation

The main goal of reversible electroporation is to permeabilize cells, while preserving their viability. The following two reversible electroporation techniques, ECT and GET, have already been developed and brought to human and veterinary medicine applications.

### 3.1. Electrochemotherapy

ECT is a local treatment for cancer, which selectively kills replicating cells. In a typical ECT session, the patient arrives, and the chosen anesthetic procedure is performed (local for the treatment of a single tumor or general for multiple or extensive tumors). Intravenous Bleomycin (BLM) is administered, and 8 min later, when the drug reaches its peak concentration in tissues, the electric pulses are delivered. For that end a specific electrode is used, carefully covering the whole tumor surface and a safety margin [23]. Thanks to its selectivity, large margins of healthy tissue can be treated to reduce the risk of relapse, without compromising its survival. The total treatment time is around 25 min and, most of the times, only one session is needed to achieve an excellent response. After the treatment, the patient returns home and waits for tumor shrinkage over the following months. 

The electric pulses used in ECT are well established, they consist of 8 monopolar square-wave pulses of 100 μs duration, with an amplitude of around 1000 V/cm, and a pulse repetition frequency, which can range from 1 to 5000 Hz [24,25]. BLM is the preferred drug, but also cisplatin (CDDP) can be used. The efficacy of BLM increases more than 1000 times, while in the case of CDDP it increases 3–4 times, providing excellent local control of the treated tumors [7,25,26,27]. The intracellular concentration of BLM determines its mechanism of action. The internalization of only a few molecules induces G2-M arrest in cells (slow mitotic cell death). Thus, tumor tissues with high cell turnover are much more susceptible than normal cells. In contrast, when millions of BLM molecules enter the cells, they act as endonucleases and induce apoptosis [28]. Moreover, BLM generates reactive oxygen species, which cause DNA damage [29]. CDDP’s mechanism of action, however, consists in the formation of intra/inter-strand cross-links within DNA, which hinder repair mechanisms and lead to apoptosis [30].

Overall, three biological mechanisms determine the antitumoral effect of ECT. First, direct cytotoxicity is obtained by increasing the concentration of chemotherapy delivered to tumor cells [9,30]. Second, two anti-vascular effects: transient vasoconstriction that entraps chemotherapy inside the tumor, and a longer-lived effect which affects the endothelial cytoskeleton compromising the barrier function of the microvascular endothelium. This delayed effect results in tumor starvation and thus contributes to cancer cell death [31,32]. Third is immune system stimulation, which is due to the release of damage-associated signals that trigger an intense stimulation of cancer immunity overcoming tumor-immune evasion [33].

ECT treatments induce a local immune response, which is mediated by different effector pathways. Briefly, after ECT, inflammatory infiltrates are seen in the tumor, with the consequent dendritic cell maturation and migration to the draining lymph node. Activation of monocytes and T lymphocytes has been described, but it may not be enough to induce a response in distant non-treated lesions, i.e., abscopal effect. The induced stimulation of the immune system by ECT is mediated by (i) the exposure of calreticulin in the cell membrane (a protein that signals the macrophages to attack the cells), (ii) the liberation of adenosine triphosphate (boosting antitumor immune response), and (iii) the liberation of high mobility group box 1 protein (which binds to the receptor of advanced glycosylation end products, a multiligand receptor, promoting inflammation). All of these are hallmarks of immunogenic cell death [34]. ECT, like IRE, elicits a local immune response, which improves treatment outcomes. Immunodeficient mice display a lesser response to ECT when compared with immunocompetent ones [35]. BLM by itself can induce immunogenic cell death via endoplasmic reticulum stress response and reactive oxygen species generation, leading to membrane translocation of endoplasmic reticulum proteins. Among these proteins are calreticulin and ERp57, which induce immunogenic apoptosis in certain tumors. BLM reactivates the antitumor immunity mediated by CD8+ T cells that is suppressed by the tumor microenvironment due to “cancer immunoediting” [36]. In addition, the therapeutic efficacy of BLM is greatly improved by the ablation of TGFβ or Tregs, both immunosuppressive, indicating the role of the immune system on the effectiveness of this drug [37]. Based on the aforementioned studies, BLM is recommended for ECT whenever possible.

An abscopal effect is a rare event in ECT. However, it may occur as seen in the case depicted in Figure 1. Sometimes, after ECT, the immune response manages to stabilize the disease for a long time, as seen in the case presented in Figure 2. It is usual to see a swelling of the tumor after the procedure, as can be seen in Figure 3. This initial growth should not be confused with a relapse, as it is an inflammatory reaction that leads to the final response.

After the publication of the standard operating procedures in 2006, and the updated procedures in 2018, the application of ECT in the case of skin malignancies has been widely accepted in human medicine [25,38]. Its indications are: (i) cutaneous metastases of any histology, which are symptomatic due to bleeding, ulceration, oozing, odor, or pain; (ii) progression of cutaneous metastases, where the development of symptoms as listed above, is expected; (iii) primary skin cancers, including recurrent tumors, where other treatment modalities (surgery, radiotherapy, and systemic therapies) have failed or are not possible; (iv) patients who are receiving systemic therapy, but where cutaneous metastases are progressing or not responding despite satisfactory response to systemic treatment in internal organs; (v) patient preference for ECT, after other treatment possibilities have been thoroughly explained to the patient.

The same indications apply for veterinary medicine. ECT is a very effective therapy without significant side effects. It can be a good option when surgery is not feasible in patients with severe comorbidities or patients of advanced age [25,39,40,41,42]. In Europe, the use of this therapy is being explored for the treatment of deep-seated tumors [43,44]. 

The situation in Latin America is quite the opposite. While veterinary electroporation-based treatments are steadily growing [45,46], in human medicine, they are barely starting as will be described later. The first human application of electroporation-based treatments is an ongoing clinical trial that uses GET for treating cervix cancer [47], and ECT is performed in few selected centers as a standard of care only in Argentina.

### 3.2. Gene Electrotransfer

GET is a delivery system for gene transfection. It consists in the injection of plasmid DNA at the target tissue, followed by the delivery of the electric pulses [48]. In contrast to ECT, pulsing parameters for GET encompass a wide variety of possibilities, as will be seen later. One of the most accepted protocols uses a combination of short and intense, followed by long and mild pulses. The first ones produce a transient permeabilization of the cell membrane, while the second ones induce the electrophoretic movement of DNA towards the electroporated cells [49,50,51]. Among gene therapy approaches, GET is one of the most efficient non-viral techniques. It has proven its efficiency in various animal models (mice, rats, rabbits, and companion animals) where different tissue-specific protocols have been set up [52,53]. In cells with defective genes, GET can be used to restore normal functions. In cancer cells, it can be used to induce the production of immunogenic proteins. Compared to viral-based gene therapy approaches, GET has appealing features as large transgenes can be transferred into target cells, the gene expression is transitory, and the risk of insertional mutations is low [54]. These properties are beneficial for vaccination purposes, or immunostimulatory strategies, based on the administration of genes encoding different cytokines [33]. 

GET-based DNA vaccination protocols are usually applied to skin or muscles. Antigen-presenting cells, such as dendritic cells, mainly involved in the immune response activation, are present in these locations [55]. In contrast, administering GET to cancer cells provokes a transient expression, as these cells divide rapidly and lose the plasmid by dilution over cell divisions. Nevertheless, this characteristic of tumor cell transfection is particularly useful for a short-term production of immunostimulatory molecules, such as anticancer cytokines [56], where the idea is to induce robust cell-mediated immunity to attack and destroy cancer cells bearing a particular antigen [33,57,58]. The common goal of most anticancer immunotherapies is to produce a cytolytic response mediated by CD8+ T cells, associated with the secretion of Th1 cytokines. However, the ideal effect would consist of coordinated activation of all effector components of the adaptive immune system, namely CD8+ T cells, CD4+ T cells, and antibodies [59,60,61]. Among strong candidates for this end is interleukin-12 (IL-12) because it enhances CD4+ T cell differentiation into Th1 cells, stimulates cytotoxic functions of CD8+ T cells, NK cells, and NKT cells by increasing IFNγ secretion, and shows anti-angiogenic properties [62].

GET using a plasmid encoding IL-12 has been tested in several preclinical studies and demonstrated its efficiency and safety on many tumor models [63,64]. The encouraging results obtained in the models led to two significant clinical trials. The first one in 2008, was a phase I study performed on human patients with metastatic melanoma, where 10% of the patients had a complete response of distant untreated lesions and 42% showed disease stabilization or partial response [65]. In 2020 the results of the phase II study performed on human patients using the same plasmid were published, where 46% of the patients showed some degree of systemic response, and 75% of the treated patients remained alive after a 6-year follow-up period [66]. These results are not only due to the IL-12 but also to the GET procedure itself, as it increases the immunogenicity of DNA vaccines as compared with DNA injection alone [67]. GET shares the immunostimulatory effects produced by all electroporation-based treatments, worth mentioning: (i) leukocyte infiltrates which can be seen after pulse delivery into the muscle, (ii) antigen-presenting and dendritic cell recruitment, (iii) moderate T CD4+ and modest T CD8+ lymphocyte recruitment, and (iv) a significant increase in local IL-1β and TNF-α release [68].

However, improving GET is not just choosing the right plasmid. Many efforts are in place through in vitro and in vivo studies to optimize GET protocols to minimize tissue injury and enhance gene transfection efficiency [69,70,71,72]. In addition, a successful strategy to increase DNA uptake and expression is the use of hyaluronidase, an enzyme capable of degrading hyaluronic acid, a component of the extracellular matrix. This allows the plasmid to get closer to the cell membrane. In this way, DNA is more available for its internalization after the pulse delivery [25,73,74]. 

Currently, among more than four thousand gene therapy-related human clinical trials, more than 30 are based on GET strategies related to immunotherapy, either for infectious diseases or cancer (www.clinicaltrials.gov).

Main features of the most used electroporation-based treatments are displayed in Table 1.

## 4. ECT and GET Immunomodulatory Effects: A Promising Combination Strategy

As previously discussed, all electroporation-based treatments induce a local immune response, making them ideal candidates to be combined. In particular, ECT is an excellent local therapy to be combined with any form of immunotherapy for achieving a systemic response.

In veterinary medicine, the first attempts consisted in injecting immunostimulatory molecules after the ECT, such as TNF-α [75], IL-2 [76], and IL-12 [77], among others. Some encouraging results were obtained, but dosing and toxicity were challenging to manage. It was shown that insufficient stimulation of the immune system might paradoxically enhance tumoral growth instead of impeding it [78,79,80]. Thus, a wrong dose may discard a promising research line, and for that reason, extreme caution should be taken to avoid this.

As a second option, the combination of ECT with checkpoint inhibitors, such as ipilimumab, pembrolizumab, or nivolumab, showed promising results linked to systemic responses in human patients [81,82,83]. However, costly immunotherapies are not an option for veterinary medicine in low and middle-income countries.

In this context, GET emerges as the natural choice for immunotherapy to be combined with ECT for various reasons. First, the same device used for ECT can be used for GET. Second, ECT users already have the know-how. Third, plasmid production can be done at a fraction of the cost of the newest immunotherapies. Finally, it can provide a more stable and sustained availability of immunomodulatory molecules in the target site. For these reasons, the combination of ECT and GET can achieve excellent results in animal patients with spontaneous tumors. In Table 2, a summary of the protocols used in veterinary patients with spontaneous tumors is presented.

In GET, a broad set of parameters seem to work fine, as they can permeabilize the membrane allowing the DNA entrance into the cells. However, not every electroporator on the market can deliver the pulses used in every protocol; special care should be taken to choose a protocol that the device being used can handle. The available devices and their capabilities will be described later.

Up to the date of the writing of this work, GET was mainly used to increase the efficacy of other therapies. Most of the patients treated were at very advanced stages of the disease, where standard treatments are known to be ineffective. Even in these unfavorable conditions, the stimulation of the immune system with GET provided an improvement in treatment outcome. Moreover, two authors reported a systemic effect using GET: one paper [85], and a congress presentation [90]. Additional benefits of including GET in the treatment protocols are reduced incidence of metastases, better local responses, and extended survival times [85,88,92,95].

When performing ECT with bleomycin, the plasmid should be administered at a different time or location than the ECT. The endonuclease activity of the bleomycin may destroy the plasmid. In addition, if the transfection is produced in a cell that will die after the first division, a lesser expression of the transgene will be obtained. 

However, many of the benefits attributed to the combination of GET and ECT have been reported for ECT alone [24,34,97,98]. Dissecting the contribution of each treatment to the local response, when GET and ECT are combined, would require a comparative study with a large number of patients, as both are highly effective. 

In several animal models, the capability of achieving a systemic response using GET alone has been demonstrated [99,100,101]. Similar results were obtained in human patients, where promising results were reported [65,66], and many clinical trials are going in this direction.

We are at an early stage of research in this matter; only treatments using IL-12 (human, feline, or canine) have been tested in dogs with spontaneous tumors, and those studies were mainly focused on safety and local response. The field is mature enough for a step forward to be taken. Future studies have to be designed to explore the systemic effect of ECT plus GET treatments in veterinary patients with spontaneous tumors.

## 5. State of the Art of Electroporation-Based Treatments in Latin America

In Latin America, several research projects on electroporation-based treatments started in the last decade, both to elucidate their fundamental aspects and to extend their applications. In particular, the group of Dr. Marshall made several publications on this topic [10,70,97,102,103,104,105,106,107,108]. The use of ECT in veterinary medicine began in 2008 in Brazil by the group of Dr. Rangel [109,110,111,112,113,114], closely followed by the group of Dr. Marshall, from Argentina, in 2009. Although the treatment had been used in human medicine in Europe since 2006, there were no devices available in Latin America for human nor veterinary medicine at that time. This lack of electroporators and also the lack of information hindered the implementation of this technology. The ECM 830 (BTX-Harvard Apparatus, Holliston, MA, USA) was the first laboratory device to be used in veterinary medicine in Latin America, with all the drawbacks that using a lab device for clinical procedures has. Electrodes had to be developed by each group, following the advice and recommendations from the literature and experts.

As more experience was gathered, in 2013, the first ECT course for veterinarians in Latin America was made available in Portuguese, directed by Dr. Rangel. Portuguese is only spoken in Brazil, while all the other Latin American countries speak Spanish. It was not until 2018, with the availability of a training course in Spanish directed by Dr. Marshall, Dr. Maglietti, and Dr. Tellado, that the technique spread to many more countries. Most of the information regarding electroporation-based treatments is in English and, for that reason, it is not accessible for most of the veterinarians in Latin America. Consequently, the availability of updated information in Spanish was highly awaited (https://vetoncologia.com/cursos/curso-electroquimioterapia-veterinaria-online/). In 2019, the International Society for Electroporation-Based Technologies and Treatments (ISEBTT) granted its support for this course, validating the imparted knowledge outside Europe and the United States. In 2020 the first edition of the Book of Electroporation-Based Technologies and Treatments was translated into Spanish by Dr. Maglietti. It includes core lecture material of the Electroporation-Based Technologies and Treatments School held in Ljubljana, Slovenia since 2003 (www.ebtt.org), and is available free online (book.ebtt.org).

The use of ECT grew at an incredible speed, with more than 20,000 patients treated in more than 150 centers by the end of 2019. Ten out of the twenty countries of Latin America have ECT: Argentina, Brazil, Chile, Colombia, Ecuador, Guatemala, México, Paraguay, Perú, and Uruguay (see Figure 4). The development of the EPV-100 device (BIOTEX SRL, Buenos Aires, Argentina) introduced an extended warranty and full customer support in Spanish, which also encouraged veterinarians to start performing ECT in their clinics. Another reason for this rapid expansion of the technology is its very low cost when compared with other oncological treatments in veterinary medicine. For instance, radiotherapy, a highly effective procedure, is only available in very few centers across Latin America, and its cost makes it prohibitive for most pet owners.

Regarding GET, only two groups initiated their trials in companion animals, the group of Dr. Marshall in Argentina, in collaboration with Dr. Signori (Italy), and the group of Dr. Rangel in Brazil, in collaboration with Prof. Cemazar (Slovenia). These trials are still ongoing, and a congress presentation with promising results, mentioned before, has been published [90].

The amount of experience and knowledge gathered made possible the organization of yearly international meetings, where veterinarians and researchers presented the results of their work. The first meeting was held in Buenos Aires, Argentina, in 2018, the second in Sao Paulo, Brazil, in 2019, and future meetings will follow. Experts from Europe and the US joined these meetings, providing an excellent opportunity to share and discuss electroporation-based treatments.

## 6. The Electroporators Available for Veterinary Medicine in Latin America

There is no such thing as the perfect electroporator, and the ideal device has not yet been built. Each device has its advantages and flaws, which veterinarians should be aware of to obtain the best possible outcome of an electroporation-based treatment. Most devices are optimized for ECT applications, where electrical parameters are well defined. In GET applications, the situation is more complicated since many options exist.

The availability of different electroporators at an affordable price in Latin America made the expansion of ECT possible. Experienced users may opt to use manual devices because this allows the designing of new electrodes as pulse parameters can be adjusted for each particular development. However, it is crucial to comply with the pulse parameters established in the Standard Operating Procedures for Electrochemotherapy to obtain the best results [25]. For instance, to deliver 1000 V/cm in a custom-built electrode with needles separated 4 mm from each other will require setting the device to deliver 400 V; in another case, with a separation of 8 mm, the device should be set to deliver 800 V. Pulse number and duration should not be changed (8 pulses of 100 µs) unless for research purposes. The frequency has no impact on the result of the treatment, ranging from 1 to 5000 Hz. Above 100 Hz, only one muscle contraction is seen, and closer to 5000 Hz, the muscle contraction is less intense [115]. For regular use in a veterinary setting, an automatic device is the best choice. Self-configured devices are safer and provide better treatment outcomes as they prevent mistakes in pulse configuration. 

Some devices deliver bipolar pulses, instead of the classical monopolar ones. The use of monopolar or bipolar pulses has no impact on ECT effectivity. However, bipolar ones may show some advantages for GET, as long as the pause between the positive and the negative deflection is minimum (symmetrical bipolar pulses), otherwise this advantage is lost [116].

The device must sustain the voltage during the pulsing procedure, but unfortunately, checking this requires an oscilloscope. The maximum current output of an electroporator is linked to its ability to avoid voltage drops. Some electrodes with long needles and tumors with lower electrical resistance will require a high electric current output for proper treatment.

Not every manufacturer is willing to ship their devices to Latin America. In Table 3, a list of electroporators that can be found in Latin America is presented. All the devices mentioned are capable of performing ECT. However, to perform GET we have to choose a protocol that our device can perform. The high-voltage followed by low-voltage pulses (HV+LV) protocols can be implemented only with the Clinivet (IGEA, Carpi, Italy) or with the BTX ECM 830 (Harvard Apparatus, Holliston, MA, USA) electroporators. Some devices have specific modes for GET like the EPV-100 (BIOTEX, Argentina), the Oncopore (BIOTEX, Argentina), and the Clinivet (IGEA, Carpi, IItaly), where the devices are self-configured. In other cases, the protocols more similar to ECT can be used. 

The Cliniporator (IGEA, Carpi, Italy) is the first electroporator approved for human use. However, it is not available in Latin America, where there is a single device approved for human use, the Oncopore (BIOTEX, Argentina).

Electrodes are an essential part of each device. They are designed to deliver an adequate electric field for optimal treatment, as long as the electrodes provided by the manufacturer are used. Different models allow the treatment of different parts of the body with more or less difficulty. Typical electrodes have needles or plates which have specific indications for their proper use that the treating professional should be aware of. Reusable electrodes are only available for veterinary medicine as human medicine only uses disposable ones. They add a plus to the efficacy of the treatment, as they will not rust (rust can isolate parts of the needle even after a single use), and they have the best sharpness for each treatment. An example of different electrodes can be seen in Figure 5.

## 7. Final Considerations and Conclusions

Human and veterinary electroporation-based treatments share many similarities in their essential aspects. However, in practice, veterinary treatments have some particularities.

Veterinary patients usually arrive at the first visit in advanced stages of the disease, mainly because of the lack of oncologists among veterinarians who can make an early diagnosis, a tendency that has been reversing in recent years. This situation reduces the success of any electroporation-based treatment, in particular ECT, as in bigger tumors a more unsatisfactory response tends to be achieved [98]. Encouraging owners to seek consultation earlier will improve the results of any treatment applied. Veterinarians cannot question their patients, and they have to rely on the answers of the owners and support them. Pets do not feel depressed because of having cancer, or because of being severely disfigured, but they are very affected when their owner is.

The most commonly treated species are canine, feline, and equine. Even in the same species, a great variety of sizes and weights are present that will require some considerations as follows.

Regarding ECT, when treating small animals, like cats and dogs using intravenous BLM, less time is required for the distribution of the drug (5 min instead of the typical 8 min). The use of CDDP is not recommended for cats. In equines, local administration of the drug is often used, as intravenous medication requires large volumes of drugs that would render the treatment extremely costly. Often, specially designed electrodes are needed for optimal treatment.

Regarding GET, despite the high similarity that some genes display across species, it is always better to use species-specific genes. This will prevent the formation of antibodies that neutralize the therapeutic protein. Moreover, using a standard electrode that permeabilizes a certain volume of tissue could be very different among different animals. For instance, it is not the same to permeabilize a volume of 8 × 8 × 10 mm in a poodle as opposed to a horse. In this case, the amount of therapeutic protein produced may be the same, but the dose per kg, and then the effect, will be different.

Exotic animals are a completely new chapter in which special care and knowledge are needed to perform a successful treatment, as standard guidelines may not be applicable [117,118,119,120]. A similar case occurs with livestock animals, where these therapies can be very useful for animals with reproductive value such as bulls. However, specific regulations apply regarding the use of chemotherapy or gene therapy in these animals, and they vary depending on the country. 

The availability of diagnostic procedures in veterinary medicine is limited, mainly due to financial constraints. Thus, performing immunohistochemistry, computed tomography scans, and magnetic resonance images is not always possible, unlike in human medicine. 

With the current advances in knowledge about cancer interaction with the immune system, we now know that cancer cells can evade, downregulate, or even benefit from the host’s immune response [79,121,122,123]. Great efforts are being made to restore the immune system’s protective function against cancer, and new immunomodulatory drugs such as pembrolizumab and ipilimumab were developed and showed promising results when combined with standard therapy [124]. These drugs have been developed since 1997 and target a wide variety of antigens. Several clinical studies in human patients have been made to demonstrate their effectiveness for different types of cancer [125]. However, their extremely high cost makes their use in low and middle-income countries very difficult in human oncology and impossible in veterinary oncology [126]. These treatments are only reserved for a wealthy sector of the society, especially in low- and middle-income countries where average salaries are usually barely enough for living. Providing adequate oncological treatment to pets is a significant economic effort. In this setting, electroporation-based therapies democratize the access to treatment for pets with cancer, as is demonstrated by the considerable growth that these therapies have had in Latin America.

An additional benefit is that a lot of experience is being gathered in the veterinary clinical setting, and it is well known that companion animals with spontaneous tumors are the best models for translational oncology [127,128,129]. The vast number of clinical trials in veterinary medicine on ECT and GET give very valuable information that can be used as preclinical validation for procedures to be used in human clinical trials. Nevertheless, in the case of GET, the path should be walked the other way around, as results in human medicine are more advanced than in veterinary medicine. So often, different animals have provided their bodies to serve as research subjects or even for learning or training purposes. Now it is time to return that favor. Researchers of human medicine should now provide their experience to veterinarians, so that they can adapt the protocols and procedures to achieve a successful immunotherapy protocol using electroporation-based treatments in companion animals. The combination of ECT plus GET can thus be the warhorse that we were waiting for.

## Figures and Tables

**Figure 1 vaccines-08-00537-f001:**
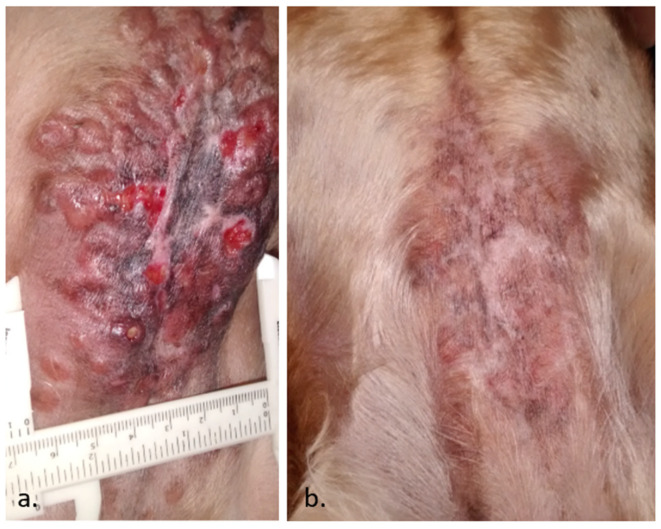
A case of a canine sebaceous gland carcinoma in a 4-year-old female cocker spaniel. In (**a**), the initial presentation, with a large lesion of 15 × 6.5 cm containing multiple nodules and ulcers. Only 20% of the tumors were treated with Electrochemotherapy (ECT) due to the large extension and spread of the disease, intending to repeat more sessions in the future. In (**b**), the patient achieved a complete response within 12 weeks of a single ECT session. Fifteen months later, the patient remains free of disease (data not published).

**Figure 2 vaccines-08-00537-f002:**
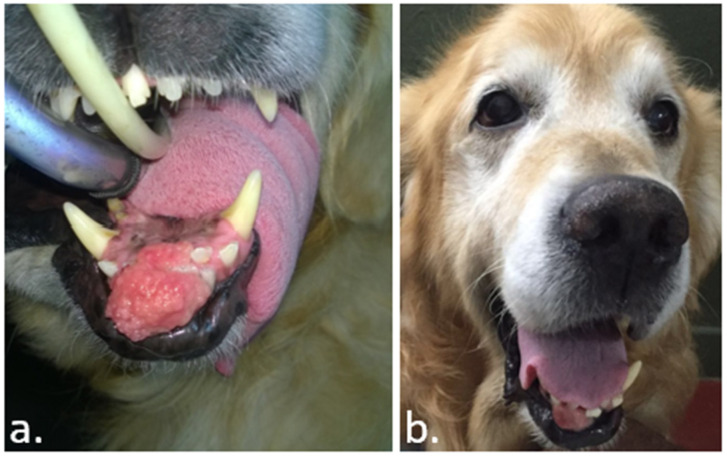
A case of a golden retriever with a fibrosarcoma, in (**a**), before the ECT. In (**b**), the patient obtained a partial response after the treatment. In contrast to the natural evolution of the disease, it remained stable without any other treatment after two years (data not published).

**Figure 3 vaccines-08-00537-f003:**
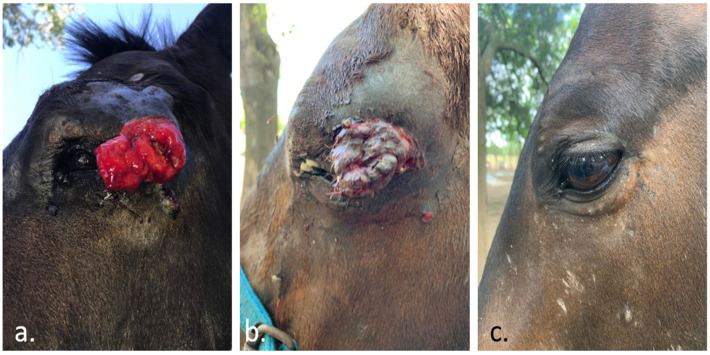
In this case, a horse with a sarcoid in the eyelid. In (**a**), the day the ECT was performed. In (**b**), thirty days after the ECT, the tumor experiences an increase in its size due to swelling. In (**c**), after a single session of ECT, a complete response was obtained. The animal remained disease-free for 400 days, and up to the date of the writing of this work (data not published).

**Figure 4 vaccines-08-00537-f004:**
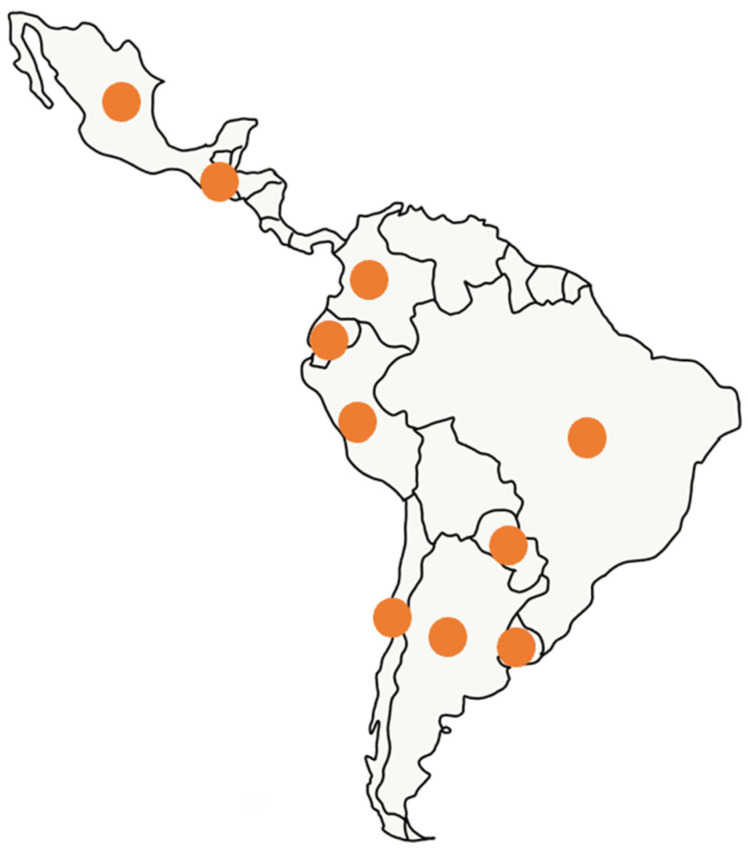
Map of Latin America. Orange spots indicate countries where ECT is performed as a standard of care in veterinary medicine.

**Figure 5 vaccines-08-00537-f005:**
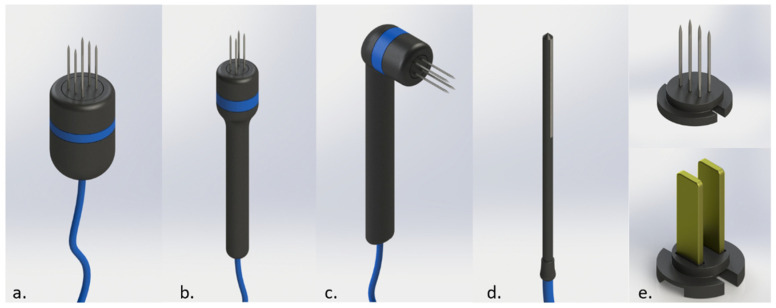
Different electrodes for ECT in veterinary medicine. In (**a**), a six-needle electrode. In (**b**), a four-needle electrode using thin-needles. In (**c**), a 90 degrees handle for the thin-needles electrode. In (**d**), an electrode for treating nasal duct. In (**e**), on top, disposable-needles, at the bottom, plates electrode that can be attached to the handles. Courtesy of BIOTEX SRL.

**Table 1 vaccines-08-00537-t001:** Main characteristics of Electrochemotherapy (ECT), Gene Electrotransfer (GET) and Irreversible Electroporation (IRE).

	ECT	GET	IRE
Spares non-dividing cells	yes	yes	no
Drugs used	bleomycin or cisplatin	plasmids	none
Electrical parameters	8 × 1000 V/cm 100 µs	varies	80–90 × 1000–30,000 V/cm 100 µs
Thermal effect	no	no	depends on the protocol
Regular use in human medicine	yes	no	yes
Regular use in veterinary medicine	yes	no	no
Uses	cancer	cancer, vaccines, gene therapy	cancer, ablation of tissues (cardiac)

**Table 2 vaccines-08-00537-t002:** Summary of research articles published using Gene electrotransfer (GET) alone or in combination with Electrochemotherapy (ECT), or other strategies for treating spontaneous tumors in canine patients. The reported species of the genes used is included in brackets.

Year	Technique	Plasmid	Pulse Parameters	Results	Author
2008	GET	IL-12 (human)	1 × 600 V/cm 100 µs + 4 × 80 V/cm 100 ms	Safety n = 6	Pavlin [84]
2008	ECT + GET	IL-12	2 × 450 V/cm 25 ms	CR: 100% n = 2 An untreated part of a lesion responded	Cutrera [85]
2010	ECT + GET	IL-12 (feline)	2 × 400 V/cm 20 ms	CR: 80% PR: 20% n = 5	Reed [86]
2011	GET + standard therapy	IL-12 (human)	1 × 1200 V/cm 100 µs + 8 × 140 V/cm 50 ms	CR: 36.4% n = 8 (11 tumors)	Pavlin [87]
2011	GET + standard therapy	IL-12 (human)	1 × 600 V/cm 100 µs + 4 × 80 V/cm 100 ms	CR: 33% n = 6	Cemazar [88]
2015	ECT + GET	IL-12 (canine)	2 × 350 V/cm 20 ms	CR: 14% PR: 36% n = 13	Cutrera [89]
2016	ECT + GET	IL-2 (canine) + IL-12 (canine)	ECT 8 × 1000 V/cm 100 µs GET 1 × 1000 V/cm + 4 × 140 V/cm 100 ms (with hyaluronidase pretreatment)	SD: 100% One systemic response n = 5	Maglietti [90]
2017	GET	IL-12 (human)	2 × 750 V/cm 50 µs + 8 × 183 V/cm 10 ms	Authors describe immunostimulatory effects. n = 9	Cicchelero [91]
2017	ECT + GET	IL-12 (human)	ECT: 8 × 1300 V/cm 100 µs GET: 1 × 1200 V/cm 100 µs + 1 × 140 V/cm 400 ms	CR: 72% n = 18	Cemazar [92]
2017	GET + ciclophosphamide	IL-12 (human)	2 × 750 V/cm 50 µs + 8 × 183 V/cm 10 ms	PD Slower tumor growth and improved well-being n = 6	Cicchelero [93]
2017	ECT + GET	IL-12 (human)	ECT: 8 × 1300 V/cm 100 µs GET: 1 × 1200 V/cm 100 µs + 1 × 140 V/cm 400 ms	Reduced neoplastic cell proliferation and induced cellular response. n = 11	Salvadori [94]
2019	ECT + GET + Surgery	IL-12 (canine)	ECT 8 × 1300 V/cm 100 µs GET 2 × 60 V/cm 150 ms	OR: 67% n = 9	Milevoj [95]
2020	GET + various treatments	IL-12	not reported	OR: 82% n = 44	Milevoj [96]

**Table 3 vaccines-08-00537-t003:** Commercially available electroporators in Latin America.

Device, Company	Pulse-Type (Square)	Max Current [A]	Automatic/Manual	Max Frequency [Hz]	Electrodes Available	Company Location
EPV-100, BIOTEX (www.biotex.com.ar)	monopolar	55	Automatic	5000	Disposable needles in three versions, plates, nasal, and urethral	Argentina
Oncopore, BIOTEX (www.biotex.com.ar)	monopolar	55	Automatic	5000	Disposable needles in three versions, plates, nasal, and urethral for human use.	Argentina
VETCP 125, VETCP (www.vetcancer.com.br)	bipolar	25	Manual	5000	Reusable needles.	Brazil
BK100, BRUNNER	monopolar	10	Automatic	1	Disposable needles	Brazil
ElectroVET EZ, LEROY BIOTECH (www.leroybiotech.com)	monopolar	25	Automatic	5000	Reusable needles and plates	France
OnkoDisruptor, BIOPULSE BIOTECH (www.onkodisruptor.com)	bipolar	5	Automatic	Not reported	Reusable needles and plates	Italy
ECM 830, BTX (www.btxonline.com)	monopolar	100	Manual	10	Not provided for vet medicine	United States
Clinivet, IGEA (www.igea.it)	monopolar	25	Automatic/Manual	5000	Reusable needles in three versions and plates	Italy

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
