# Peer review of "Electroporation as the Immunotherapy Strategy for Cancer in Veterinary Medicine: State of the Art in Latin America"

_vaccines, 2020, doi:10.3390/vaccines8030537_

Round 1

Reviewer 1 Report

Manuscript submitted by Maglietti et. al., entitled as "Electroporation as the immunotherapy strategy in veterinary medicine:state of the art in Latin America" provide an interesting overview of the current practices in veterinary medicine regarding use of electroporation as a tool in immunotherapy. Authors have introduced various methods used in this kind of therapy, with an emphasis on treatment of cancer. 

Authors have several important points to make, but due to lack of a storytelling freeflow, and at times information overload without giving proper context, manuscript will not be attractive to readers. Throughout manuscript, authors need to follow a theme and organization and create a logical and coherent flow of the information. In the present format it is very confusing and readers may easily loose interest. 

Introduction is disorganized. Introduction need to provide scope and rationale of the manuscript. Why and how things are different in Latin American than rest of the world? Authors should discuss "electroporation", chemotherapy in general, and electrochemotherapy in particular. List of monoclonal antibodies is not useful here. Authors may like to stick to the concept of EP in cancer therapy. This will make article more focussed and give a better logic. 

Regarding EP, authors need to describe why EP is not prevalent among human medicine. It should be the first section. Give its principle, indications, and usage in cancer therapy. 

At places, several terms are randonly introduced without providing any context. E.g. Bleomycin or cisplatin in line 100. There is no connection with previous paragraph, and entire paragraph does not discuss EP etc. So, this information is out of context and does not fit with the theme of the manuscript. This is an issue throughout the manuscript. 

Similarly, Figs 1-3, are lacking context. Does it justify use of EP? What drugs were introudcted with ECT? How treatment was performed? Overall, what is the takehome message here? Each and every section should have an aim and a take home message. 

Fig 4 should be moved as Fig 1, along with introduction. Also, it would be nice to indicate what countries make Latin America. A geopolitical map, showing name of different countries will be helpful. In some countries ECT is not a "standard of care in veterinary medicine", authors should discuss it. 

For Fig 5, how does this knowledge of various kinds of electroporators is helpful to the readers? Again context is missing. 

Overall, manuscript in present format is just a compilation of relevant information. It should be rearranged, paraphrased so that a logical flow could be established which may drive curiosity and enthusiam among readers.

Author Response

The responses to the reviewers are inline in red.

Manuscript submitted by Maglietti et. al., entitled as "Electroporation as the immunotherapy strategy in veterinary medicine:state of the art in Latin America" provide an interesting overview of the current practices in veterinary medicine regarding use of electroporation as a tool in immunotherapy. Authors have introduced various methods used in this kind of therapy, with an emphasis on treatment of cancer. 

We thank very much reviewer #1 for taking the time of reviewing the manuscript and providing a very interesting correction that hopefully improved the manuscript.

Authors have several important points to make, but due to lack of a storytelling freeflow, and at times information overload without giving proper context, manuscript will not be attractive to readers. Throughout manuscript, authors need to follow a theme and organization and create a logical and coherent flow of the information. In the present format it is very confusing and readers may easily loose interest. 

The manuscript has been thoroughly revised, and the English have been corrected. The entire sections had been rewritten following the referee´s suggestions.

Introduction is disorganized. Introduction need to provide scope and rationale of the manuscript. Why and how things are different in Latin American than rest of the world? Authors should discuss "electroporation", chemotherapy in general, and electrochemotherapy in particular.

Following the referee´s suggestion, the introduction has been reorganized and in particular, electroporation, gene electrotransfer and electrochemotherapy have been discussed.

List of monoclonal antibodies is not useful here.

The list has been removed.

Authors may like to stick to the concept of EP in cancer therapy. This will make article more focussed and give a better logic. 

We agree with the reviewer. The title has been changed to reflect the focus of the manuscript in cancer. The whole text has been revised to focus mainly on cancer applications.

Regarding EP, authors need to describe why EP is not prevalent among human medicine. It should be the first section. Give its principle, indications, and usage in cancer therapy. 

We added the suggestions for a better understanding of the technology. Also, as suggested the indications, and use in cancer therapy has been added.

At places, several terms are randonly introduced without providing any context. E.g. Bleomycin or cisplatin in line 100. There is no connection with previous paragraph, and entire paragraph does not discuss EP etc. So, this information is out of context and does not fit with the theme of the manuscript. This is an issue throughout the manuscript. No se entendió lo que es la ECT.

Bleomycin and cisplatin are the only drugs that are approved for electrochemotherapy. This concept has been extended in the text to make it clear to the reader.

Similarly, Figs 1-3, are lacking context. Does it justify use of EP? What drugs were introudcted with ECT? How treatment was performed? Overall, what is the takehome message here? Each and every section should have an aim and a take home message. 

The introduction to Figures one to three in the text has been explained in more detail. They display the immune system response that the treatment can produce. Thou this effect is rare, it could be enhanced by the combination of ECT with GET.

Fig 4 should be moved as Fig 1, along with introduction. Also, it would be nice to indicate what countries make Latin America. A geopolitical map, showing name of different countries will be helpful. In some countries ECT is not a "standard of care in veterinary medicine", authors should discuss it. 

We thank the reviewer for the observation, but when we added the names to each country, the graph was not so clear. For that reason, we decided to exclude the names in the figure. Nevertheless, the list of countries that have electrochemotherapy is included in the text. When we write as a “standard of care”, we mean “not under a trial”.

For Fig 5, how does this knowledge of various kinds of electroporators is helpful to the readers? Again context is missing. 

To choose the proper GET protocol, the owner should know the capabilities of their devices. It may be surprising, but most of the users of ECT lack the knowledge of these aspects. For that reason, many times a very low GET effectivity is obtained.

Overall, manuscript in present format is just a compilation of relevant information. It should be rearranged, paraphrased so that a logical flow could be established which may drive curiosity and enthusiam among readers.

We hope to satisfy the reviewer with this new version that followed very close to his advice. We believe that the information presented is not only interesting but also very useful for electroporation-based treatment users.

Reviewer 2 Report

ECT and GET are interesting alternative approaches for treating cancer and transfecting nucleic acids in vivo. These strategies, which have shown efficacy in humans, are cheaper than current immunotherapies used in human medicine. Therefore, they are a very attractive option in veterinary medicine. In this review the author discuss ECT and GET and their application and efficacy in veterinary medicine. The discussion of the available EP devices is interesting for the reader.

However, in the past and also recently this topic has been reviewed. Especially the group of Enrico Spugnini made several reviews. The latest was entitled: “Electrochemotherapy in Veterinary Oncology: State-of-the-Art and Perspectives” (Vet Clin North Am Small Anim Pract 2019 Sep;49(5):967-979; doi: 10.1016/j.cvsm.2019.04.006). Another review was published in 2016 by Impellizeri et al. (Vet J. 2016 Nov;217:18-25. doi: 10.1016/j.tvjl.2016.05.015). These reviews should at least be mentioned in this overview.

The section on ECT is not covering all studies for example the following papers are not included: e.g. (1) Open Vet J. 2019 Oct;9(3):269-272. doi: 10.4314/ovj.v9i3.13. Epub 2019 Sep 19; (2) Open Vet J. 2019 Apr;9(1):88-93. doi: 10.4314/ovj.v9i1.15. Epub 2019 Mar 25, (3) J Vet Intern Med. 2015 Sep-Oct;29(5):1368-75. doi: 10.1111/jvim.13586. Epub 2015 Jul 20 , (4) J Vet Intern Med. 2011 Mar-Apr;25(2):407-11. doi: 10.1111/j.1939-1676.2011.0678.x. Epub 2011 Feb 11.

Table 1 gives an overview of studies that used GET alone or in combination with ECT. This is a very informative table. Could the authors make a similar table for studies involving only ECT. Additionally, there are other studies that are not mentioned in the review and in table 1. For example Vet Comp Oncol. 2017 Dec;15(4):1187-1205. doi: 10.1111/vco.12255. Epub 2016 Aug 9.

Considering ref 92 in table 1: These authors reported PD with a slower tumor growth and improved well-being. Can this be added.

ECT is exclusively used in an oncologic context, while the applications of GET are much broader. In this review only the oncological applications of GET in veterinary medicine are discussed. The authors should broaden this and also discuss non-oncological application like e.g. vaccination (see e.g. doi: 10.18632/oncotarget.26927 and doi: 10.18632/oncotarget.7265).

Considering the combination of EGT and ECT. The author should also discuss whether EGT should be given before ECT or simultaneously. As ECT is killing cells it is to my opinion better to give first EGT and after e.g. one or more days administer the ECT. For an effective GET we need living cells and ECT may rapidly kill the transfected cells when given simultaneously.

Lines 313-316: when reading this I have the impression that these meeting stopped. Is this correct?

Figure 4: this one can be deleted. Is not very informative

Considering section 7. (1) Can the authors clearly mention whether there are certificated electroporators for veterinary use? (2) The Cliniporator of IGEA is also approved for human used, isn’t? Can the authors check this and add this device to line 366. (3) line 356: this sentence is not clear for me. What do the author mean with stained/stain? Do they mean coated? (4) To my knowledge the Clinivet (line 363, table 2) is not on the market anymore. (5) We also have a ECM830 BTX. However, this device is designed for in vitro electroporation. For in vivo electroporation the AgilePulse of BTX is used. Can the author adapt this in table 2.

When the authors write “patients” it is not always clear whether they indicate pets or humans.

The term “plaque electrode” is used many times. However, the correct English term is “plate electrode”.

There are some grammar and spelling errors. It would be interesting that a native speaker is checking this review.

Author Response

The responses to the reviewers are inline in red.

ECT and GET are interesting alternative approaches for treating cancer and transfecting nucleic acids in vivo. These strategies, which have shown efficacy in humans, are cheaper than current immunotherapies used in human medicine. Therefore, they are a very attractive option in veterinary medicine. In this review the author discuss ECT and GET and their application and efficacy in veterinary medicine. The discussion of the available EP devices is interesting for the reader.

We would like to thank Reviewer two, for taking the time to read the manuscript and provide useful and valuable comments.

However, in the past and also recently this topic has been reviewed. Especially the group of Enrico Spugnini made several reviews. The latest was entitled: “Electrochemotherapy in Veterinary Oncology: State-of-the-Art and Perspectives” (Vet Clin North Am Small Anim Pract 2019 Sep;49(5):967-979; doi: 10.1016/j.cvsm.2019.04.006). Another review was published in 2016 by Impellizeri et al. (Vet J. 2016 Nov;217:18-25. doi: 10.1016/j.tvjl.2016.05.015). These reviews should at least be mentioned in this overview.

The section on ECT is not covering all studies for example the following papers are not included: e.g. (1) Open Vet J. 2019 Oct;9(3):269-272. doi: 10.4314/ovj.v9i3.13. Epub 2019 Sep 19; (2) Open Vet J. 2019 Apr;9(1):88-93. doi: 10.4314/ovj.v9i1.15. Epub 2019 Mar 25, (3) J Vet Intern Med. 2015 Sep-Oct;29(5):1368-75. doi: 10.1111/jvim.13586. Epub 2015 Jul 20 , (4) J Vet Intern Med. 2011 Mar-Apr;25(2):407-11. doi: 10.1111/j.1939-1676.2011.0678.x. Epub 2011 Feb 11.

We thank the reviewer not only for taking the time for mentioning these papers but also for providing the doi’s that greatly facilitated their inclusion in the present manuscript.

Table 1 gives an overview of studies that used GET alone or in combination with ECT. This is a very informative table. Could the authors make a similar table for studies involving only ECT.

We focused this work on GET and its combination with ECT, rather than in ECT alone. We intentionally excluded ECT studies in vet medicine as there are already very good reviews on the subject. For instance, the ones that the reviewer mentioned previously. Besides, the list would have been very long and might be out of the scope of the present work.

Additionally, there are other studies that are not mentioned in the review and in table 1. For example Vet Comp Oncol. 2017 Dec;15(4):1187-1205. doi: 10.1111/vco.12255. Epub 2016 Aug 9.

Thank you for the observation, the above-mentioned paper has been added. Some papers were not added because either they treat induced tumors in canine patients, or they use laboratory animals. In this review, we focus on spontaneous tumors in companion animals.

Considering ref 92 in table 1: These authors reported PD with a slower tumor growth and improved well-being. Can this be added.

This observation was added to the manuscript.

ECT is exclusively used in an oncologic context, while the applications of GET are much broader. In this review only the oncological applications of GET in veterinary medicine are discussed. The authors should broaden this and also discuss non-oncological application like e.g. vaccination (see e.g. doi: 10.18632/oncotarget.26927 and doi: 10.18632/oncotarget.7265).

We focus this work mainly on the applications of GET and its combination with ECT in oncology. Reflecting on this fact, we changed the title and the orientation of the manuscript. A review of the vaccination strategies for infectious diseases using GET should have a specific review. Nevertheless, the studies mentioned are in laboratory animals (mice and dogs), and for that reason, they would have been excluded.

Considering the combination of EGT and ECT. The author should also discuss whether EGT should be given before ECT or simultaneously. As ECT is killing cells it is to my opinion better to give first EGT and after e.g. one or more days administer the ECT. For an effective GET we need living cells and ECT may rapidly kill the transfected cells when given simultaneously.

We agree with the reviewer; also we consider that it is better to apply GET and ECT at different sites, as bleomycin can cut the plasmid and thus reduce the expression. We added this to the manuscript.

Lines 313-316: when reading this I have the impression that these meeting stopped. Is this correct?

Only because of the COVID pandemic. It was organized two times, one in Buenos Aires, Argentina (2018), and the other one in Sao Paulo, Brazil (2019). It was supposed to be held in Buenos Aires in November 2020. Hopefully, after the pandemic, the events will resume.

Figure 4: this one can be deleted. Is not very informative

Probably for the reader familiar with Latin America is redundant, but one of the other reviewers found it very useful to picture the locations. For that reason, we decided to keep the image.

Considering section 7. (1) Can the authors clearly mention whether there are certificated electroporators for veterinary use? (2) The Cliniporator of IGEA is also approved for human used, isn’t? Can the authors check this and add this device to line 366. (3) line 356: this sentence is not clear for me. What do the author mean with stained/stain? Do they mean coated? (4) To my knowledge the Clinivet (line 363, table 2) is not on the market anymore. (5) We also have a ECM830 BTX. However, this device is designed for in vitro electroporation. For in vivo electroporation the AgilePulse of BTX is used. Can the author adapt this in table 2.

  1. In Latin America, there is no certification for veterinary devices, only general “electrical safety” that any electronic device in the market must comply with. All the listed devices comply with this requirement.
  2. The Cliniporator now is mentioned in the text, but it was not added to the list, the reason for not including it is that it cannot be bought in Latin America.
  3. Reusable electrodes have the problem of getting rusty, even after the first use (though they are made from “stainless steel). These rust stains difficult electric current circulation, making the electric field erratic, and not uniform. For this reason, we believe that in veterinary medicine, disposable electrodes should be used, as in human medicine. We extended this explanation in the text for the sake of clarity.
  4. The ECM830 is an incredibly robust device that we use very often for laboratory use. It is intended for laboratory rather than veterinary use. For this reason, it is not provided with adequate electrodes. The AgilePulse, also a very powerful and versatile device, is intended for GET treatment of laboratory animals, and it has the same problem. They are not intended for clinical veterinary use, and for that reason, they were not included in the table. The Clinivet can be bought used from previous owners, being an excellent device, we decided to keep it in the list.

When the authors write “patients” it is not always clear whether they indicate pets or humans.

This was corrected through the manuscript.

The term “plaque electrode” is used many times. However, the correct English term is “plate electrode”.

We thank the reviewer for this correction. The name of the electrodes was corrected.

There are some grammar and spelling errors. It would be interesting that a native speaker is checking this review.

Thank you for this suggestion. The manuscript was thoroughly revised and corrected.

Reviewer 3 Report

The manuscript is informative in terms of technological details of ECT and GET for veterinarians, but it is not so useful to the general reader.

Authors focused on the state in Latin America. But, the analyses seem not enough.

Figure 4 indicates Latin America countries where ECT is performed as a standard care in veterinary medicine. How do the authors know the fact? Is it the results of scrutiny of veterinary treatment methods in these countries?

Table 2 shows commercially available electroporators in Lain America. The table seems nonsense as any other electroporators are available in Latin America. The meaning of “available in Latin America” is vague and not scientific. Does “origin” in the table mean the location of the company?

Line 21. Why is electrochemotherapy the most “notorious” application? It needs the explanation.

English should be brushed up through the manuscript. e.g., the sentences of line 399, 400 are grammatically incorrect.

Some references are not appropriately cited. e.g., #39 is not appropriate for the reference of the ablation of TGFb and /or Tregs. (Line 131).

Author Response

The responses to the reviewers are inline in red.

The manuscript is informative in terms of technological details of ECT and GET for veterinarians, but it is not so useful to the general reader.

We thank the reviewer for the comments. We agree that ECT or GET users may find a lot of valuable information, but the general reader may find it confusing. For that reason, we extended the explanations and introductions to the different topics to put the general reader into perspective.

Authors focused on the state in Latin America. But, the analyses seem not enough.

Perhaps the reviewer can suggest any details that should be included in the manuscript.

Figure 4 indicates Latin America countries where ECT is performed as a standard care in veterinary medicine. How do the authors know the fact? Is it the results of scrutiny of veterinary treatment methods in these countries?

Yes, many of the ECT users in Latin America either studied the technique in courses in Argentina (organized by us) or Brazil organized by Dr. Rangel, a close collaborator. These are the only two courses in the region. The electroporation community in Latin America is still rather small, and we know each other. Besides, we meet every year at the Latin American Congress and every two years at the World Congress. 

Table 2 shows commercially available electroporators in Lain America. The table seems nonsense as any other electroporators are available in Latin America. The meaning of “available in Latin America” is vague and not scientific. Does “origin” in the table mean the location of the company?

Unfortunately, it is not possible to buy any device from Latin America. Not every manufacturer will ship it or even sell it. Customs restrictions, taxes, and other issues discourage many manufacturers from trying to sell in the region. However, the list shows the ones that do sell and ship their devices to Latin America. The company location is displayed in the table.

Line 21. Why is electrochemotherapy the most “notorious” application? It needs the explanation.

Because since its inception its use increased very fast both in human and in veterinary medicine. Electrochemotherapy is the reason why electroporation is now a term widely known in the medical community. We rephrased that sentence in the abstract.

English should be brushed up through the manuscript. e.g., the sentences of line 399, 400 are grammatically incorrect.

We thoroughly revised the manuscript and corrected the English.

Some references are not appropriately cited. e.g., #39 is not appropriate for the reference of the ablation of TGFb and /or Tregs. (Line 131).

We agree with the reviewer and decided to change the reference to:

Bugaut, H., Bruchard, M., Berger, H., Derangère, V., Odoul, L., Euvrard, R., ... & Apetoh, L. (2013). Bleomycin exerts ambivalent antitumor immune effect by triggering both immunogenic cell death and proliferation of regulatory T cells. PLoS One8(6), e65181.

Round 2

Reviewer 1 Report

Revised manuscript has greatly improved. Kudos to the hard work by authors. There are few more gaps remaining, as detailed below.

  1. Authors have provided a great detail on use of electroporation in human medicine as well. They may like to highlight it in the title.
  2. Authors are using many abbreviations. It will be nice to have a list of all the abbreviations, to help readers navigate through the article better.
  3. Authors should use a more consistent terminology to distinguish veterinary practice from human medicinal practices. E.g. terms like clinical practice (which is both for human and vet) in Line 79, and human medicine in line 40 and in most of other places.
  4. Authors may like to create a tabular form of comparison among IRE vs REP, ECT vs GET etc. It will be helpful to summarize striking features of these different but quite similar technologies.
  5. Statement in line 225 is not scientific and should be removed. Similarly, line 223, "thousands of...." is little hyped, thousands ranges from 1000 to 99,000.
  6. At places, it is not clear if authors are discussing results from human or animal work. Authors should be more explicit.
  7. In line 293, authors can include a link to the training course, as they have provided for the book subsequently. It will be helpful to several practitioners.
  8. Table 2, All pulse is square, so this word can be put in title {Pulse-type (square)}. Units for Current and Frequency could also be included in title. A web URL (google has tiny web URL) will be very useful.
  9. Article has mainly focused on companion animals (dog, cats, and horse). Authors may also comment on the usage of these devices in livestock animals, it will be worthy for bulls etc.

Minor Comments:

Some typos need attention

  1. Line 42: "i.e. in Latin America" should be "for example ..."
  2. Line 56: "though is less effective."
  3. Line 137, it is a conclusive statement. Authors may like to paraphrase it, that "based on aforementioned studies, BLM is recommended...."
  4. Line 304, countries name should be arranged in alphabetical order.

Author Response

The responses to the reviewer are inline in red. 

Comments and Suggestions for Authors

Revised manuscript has greatly improved. Kudos to the hard work by authors. There are few more gaps remaining, as detailed below.

Thank you very much for reading again the manuscript.

  1. Authors have provided a great detail on use of electroporation in human medicine as well. They may like to highlight it in the title.

As the work focuses on results in Latin America, and as there are no publications about electroporation-based treatments in human patients in the region yet, we prefer not to include it in the title.

  1. Authors are using many abbreviations. It will be nice to have a list of all the abbreviations, to help readers navigate through the article better.

 A list of abbreviations was added.

  1. Authors should use a more consistent terminology to distinguish veterinary practice from human medicinal practices. E.g. terms like clinical practice (which is both for human and vet) in Line 79, and human medicine in line 40 and in most of other places.

A clear distinction between veterinary practice and human medicinal practice was made in all places where it was considered necessary.

  1. Authors may like to create a tabular form of comparison among IRE vs REP, ECT vs GET etc. It will be helpful to summarize striking features of these different but quite similar technologies.

A table was added to the manuscript and the main features of these technologies summarized.

  1. Statement in line 225 is not scientific and should be removed. Similarly, line 223, "thousands of...." is little hyped, thousands ranges from 1000 to 99,000.

Line 223 was corrected and now reads “More than four thousand…”. The statement in line 225 was removed. 

  1. At places, it is not clear if authors are discussing results from human or animal work. Authors should be more explicit.

Whenever results from human or animal work were discussed, this was explicitly clarified.

In line 293, authors can include a link to the training course, as they have provided for the book subsequently. It will be helpful to several practitioners.

We thank the reviewer for this suggestion. A link to the course was added.

  1. Table 2, All pulse is square, so this word can be put in title {Pulse-type (square)}. Units for Current and Frequency could also be included in title. A web URL (google has tiny web URL) will be very useful.

In Table 2 “Pulse type (square)”, units for current and frequency were added to the title. Also, the URL of each manufacturer was added.

  1. Article has mainly focused on companion animals (dog, cats, and horse). Authors may also comment on the usage of these devices in livestock animals, it will be worthy for bulls etc.

We added the following:

"Exotic animals are a completely new chapter in which special care and knowledge are needed to perform a successful treatment, as standard guidelines may not be applicable[117–120]. A similar case occurs with livestock animals, where these therapies can be very useful for animals with reproductive value such as bulls. However, specific regulations apply regarding the use of chemotherapy or gene therapy in these animals, and they vary depending on the country."

Minor Comments:

Some typos need attention

  1. Line 42: "i.e. in Latin America" should be "for example ..."
  2. Line 56: "though is less effective."
  3. Line 137, it is a conclusive statement. Authors may like to paraphrase it, that "based on aforementioned studies, BLM is recommended...."
  4. Line 304, countries name should be arranged in alphabetical order.

The typos mentioned lines 42, 56, 137, and 304 were corrected.

Reviewer 2 Report

The authors revised the review adequately.

Author Response

We thank the reviewer very much for the revision of the manuscript.

Reviewer 3 Report

Authors revised the manuscript thoroughly. After the revision, however, the manuscript is still hard to read through. The story is not so clear. And another reason is the English. There are many typographical and grammatical errors through the manuscript. Some examples are shown below.

ECT (line 42), GET (line 50), and IRE (line 59), EP (line 87) should be spelled out when they are used at the first time in text, not in Abstract.

Line 56. Is “thou” “though”?  Line 76. Is “ablative” correct?  Line 397. “puddle” should be “poodle”.

Reference #86 Reed et al. has been retracted.

Table 2. I cannot reach the company information through the internet. Is “Biotex” a company whose headquarter is located in Argentina? According to a website, a company named Biotex is located in Houston, Texas, USA. “Oncopore” is the trademark of a short anticancer peptide according to the internet. I cannot reach to VETCP.

I recommend to brush up the English by a native speaker and recheck the content carefully. 

Author Response

The response to the reviewer's comments is included inline in red.

Authors revised the manuscript thoroughly. After the revision, however, the manuscript is still hard to read through. The story is not so clear. And another reason is the English. There are many typographical and grammatical errors through the manuscript. Some examples are shown below.

We would like to thank the reviewer again for his thorough revision of the manuscript. The manuscript was carefully revised and many typos were corrected. 

ECT (line 42), GET (line 50), and IRE (line 59), EP (line 87) should be spelled out when they are used at the first time in text, not in Abstract.

ECT, GET, and IRE are spelled out when they appear for the first time in the abstract, in the manuscript, and in the figures. Also, an abbreviations list was added.

Line 56. Is “thou” “though”?  Line 76. Is “ablative” correct?  Line 397. “puddle” should be “poodle”.

Typos in lines 56 (though), 76 (ablative) and line 397 (poodle) were corrected.

Reference #86 Reed et al. has been retracted.

The correct version was cited (https://www.nature.com/articles/cgt201059).

Table 2. I cannot reach the company information through the internet. Is “Biotex” a company whose headquarter is located in Argentina? According to a website, a company named Biotex is located in Houston, Texas, USA. “Oncopore” is the trademark of a short anticancer peptide according to the internet. I cannot reach to VETCP.

Biotex headquarters is located in Argentina, its web page was added to the table. The webpage of the other manufacturers was also added.

I recommend to brush up the English by a native speaker and recheck the content carefully. 

We have proofread the manuscript again, and now, experienced scholarly writers have edited this manuscript and are confident about the English.